Genetic data of museum specimens allow for inferring evolutionary history of the cosmopolitan genus Sirthenea (Heteroptera: Reduviidae)

Chłond Dominik 1 dominik.chlond@us.edu.pl
http://orcid.org/0000-0003-0308-3991 Sawka-Gądek Natalia 2
Żyła Dagmara 3 4
1 Faculty of Biology and Environmental Protection, Department of Zoology, University of Silesia , Katowice , Poland
2 Institute of Systematic and Evolution of Animals, Polish Academy of Science , Kraków , Poland
3 Department of Ecology, Evolution, & Organismal Biology, Iowa State University , Ames, IA , USA
4 Department of Invertebrate Zoology and Parasitology, University of Gdańsk , Gdańsk , Poland
Gillespie Joseph
Electronic publication date: 2019 Apr 10
Publication date: 2019
Volume: 7
Electronic Location ID: e6640
Received 2019 Jan 30; Accepted 2019 Feb 18
Copyright: © 2019 Chłond et al.
Copyright year: 2019
Copyright holder: Chłond et al.
License: This is an open access article distributed under the terms of the Creative Commons Attribution License, which permits unrestricted use, distribution, reproduction and adaptation in any medium and for any purpose provided that it is properly attributed. For attribution, the original author(s), title, publication source (PeerJ) and either DOI or URL of the article must be cited.
License URL: https://creativecommons.org/licenses/by/4.0/

Keywords: Taxonomy, Morphology, Evolution, Molecular clock, Biogeography, Phylogeny, Total evidence

Funding: Research project of National Science Centre no DEC-2017/01/X/NZ8/00843 EU Synthesys Projects AT-TAF-2928, NL-TAF-3392, BE-TAF-5548, DE-TAF-516, SE-TAF-876, AT-TAF-5713 This work was prepared by the support of research project of National Science Centre no. DEC-2017/01/X/NZ8/00843. Dominik Chłond’s visits to NHMW, RMCA, ZMHB, RMNH and NHRS were supported by the EU Synthesys Projects: AT-TAF-2928, NL-TAF-3392, BE-TAF-5548, DE-TAF-516, SE-TAF-876, AT-TAF-5713. The funders had no role in study design, data collection and analysis, decision to publish, or preparation of the manuscript.

==============================
Among the 30 known genera within subfamily Peiratinae, only the genus Sirthenea has a cosmopolitan distribution. The results of our studies are the first comprehensive analysis concerning one of the representatives of mentioned subfamily based on joint phylogenetic analyses of molecular and morphological data as well as molecular dating. A total of 32 species were included into the dataset with all known species of the genus Sirthenea. Material of over 400 dry specimens was examined for the morphological part of this study. The cosmopolitan distribution of Sirthenea and the inaccessibility of specimens preserved in alcohol required the extraction of DNA from the dried skeletal muscles of specimens deposited in 24 entomological collections. The oldest specimens used for the successful extraction and sequencing were collected more than 120 years ago in India. We performed Bayesian Inference analyses of molecular and morphological data separately, as well as combined analysis. The molecular and morphological data obtained during our research verify the correlation of the divergence dates of all known Sirthenea species. Results of the relaxed molecular clock analysis of the molecular data show that, the genus Sirthenea started diverging in the Late Cretaceous into two clades, which subsequently began to branch off in the Paleocene. Our results of phylogenetic analyses suggest that the fossula spongiosa and its development could be one of the most important morphological characters in the evolution of the genus, most likely associated with the ecological niche inhabited by Sirthenea representatives. Confirmation of the results obtained in our studies is the reconciliation of the evolutionary history of Sirthenea with the biogeographical processes that have shaped current global distribution of the genus.

Introduction

Reduviidae, also known as assassin bugs, with ca. 6,800 described species, are one of the largest and most morphologically diverse families of Heteroptera, currently subdivided into 25 subfamilies (Zhang et al., 2016a; Weirauch et al., 2014). Among these, the subfamily Peiratinae is known from 30 described genera (Putshkov & Putshkov, 1986–1989; Maldonado Capriles, 1990; Coscarón, 1995, 1996, 1997a, 2002; Coscarón & Linnavuori, 2007; Coscarón & Morrone, 1995; Gil-Santana & Costa, 2003; Cai & Taylor, 2006). The distribution of Peiratinae has been the subject of many studies (Morrone & Coscarón, 1996, 1998; Coscarón, 2017; Coscarón & Carpintero, 1994; Coscarón & Morrone, 1995, 1997; Coscarón, 1983a, 1983b, 1984, 1986, 1994, 1996, 1997b, 2002), and it has recently been comprehensively revised (Chłond & Bugaj-Nawrocka, 2015). In addition, the above-mentioned studies show that we can clearly distinguish groups of genera with a distribution limited to specific zoogeographical regions. Among all of the known Peiratinae, only the genus Sirthenea Spinola, 1837 has a cosmopolitan distribution, which is not a frequent phenomenon within Reduviidae (Maldonado Capriles, 1990). Therefore, Sirthenea is of considerable interest as a potential model group for a wide range of studies spanning several fields such as evolutionary biology, systematics, biogeography and ecology. Our previous study showed that representatives of Sirthenea are ground-dwelling, non-specialized predators that prey on other insects on the ground in microhabitats that are mainly located in low-lying areas in tropical and temperate climates (Chłond & Bugaj-Nawrocka, 2015), which is consistent with the observations of other authors (Willemse, 1985). During the almost two centuries since the description of the first Sirthenea species as Reduvius carinatus by Spinola in 1837, the number of known species has increased to 41 (Maldonado Capriles, 1990; Cai & Tomokuni, 2004; Chłond, 2008a, 2008b), which was largely due to the high degree of variability associated with the wide distribution of selected species. The combination of the above-mentioned factors, as well as the existence of geographical barriers, in some cases enable allopatric speciation (Chłond, Bugaj-Nawrocka & Sawka-Gądek, 2017). In turn, this has resulted in a large number of described species, which are often only synonymous with geographical forms that are characterized by a high degree of polychromatism (Chłond, 2018). However, at the end of the 20th century, the number of known species of Sirthenea changed as the New World representatives were revised (Willemse, 1985), which increased the number of known Neotropical species from nine to 13. Conversely, the revision of the Old World species (Chłond, 2018) reduced the number of known species from the Afrotropical, Oriental, Palearctic and Australian zoogeographical regions by half from 28 to 14 species (including descriptions of two new species). Even though the taxonomy of the genus Sirthenea is rather well understood, no attempt has been made to study the phylogenetic relationships within the genus.

For many insect groups, the standard collecting of high-quality DNA material only began a few years ago, and therefore, DNA-grade samples for many taxa that are critical for various phylogenetic analyses are still not available. Due to the inaccessibility of specimens that had been preserved in alcohol, we used the dried skeletal muscles of specimens that belong to all of the known species of Sirthenea deposited in 24 entomological collections to obtain the molecular data for our analysis. The oldest individuals that were used for the successful extraction and sequencing belong to a newly described species, S. kali Chłond, 2018, which was collected as long ago as in 1898 in Northeast India (West Bengal). The usefulness of our method and the possibility of obtaining DNA from museum specimens of various ages that had been stored in different museum conditions were previously reported in Chłond, Bugaj-Nawrocka & Sawka-Gądek (2017) and are further confirmed here.

Our results provide a hypothesis for understanding the evolutionary history of the genus, which is supported by the evolution of the ecological niches that are occupied by representatives of Sirthenea (Couvreur & Baker, 2013; Chłond & Bugaj-Nawrocka, 2015). Moreover, all of the information that was obtained during previous studies (Chłond & Bugaj-Nawrocka, 2015; Chłond, Bugaj-Nawrocka & Sawka-Gądek, 2017; Chłond, 2018) combined with the currently obtained data, which can be linked to geological time, not only provide valuable insight into the evolutionary history of the Sirthenea, but also allow for understanding the diversity (27 species) of this cosmopolitan genus. In this study, we aimed to investigate the phylogenetic relationships of Sirthenea based on morphological and molecular data, and to estimate the divergence times within all of the known species of Sirthenea.

Material and Methods

Molecular data

Taxon sampling

The ingroup sample consisted of 27 individuals representing the genus Sirthenea. Five species were selected as the outgroup taxa for the analysis. The collection data of the specimens and their GenBank accession numbers are provided in Table S1.

DNA extraction, amplification and sequencing

DNA was extracted from the dried skeletal muscles of each specimen. Genomic DNA was isolated without modifying the protocol using the GeneMATRIX Bio-Trace kit (EURx, Gdańsk, Poland). To elute the purified DNA, we applied 50 μL of an Elution Buffer onto the silica membrane. To amplify a fragment of the mitochondrial cytochrome c oxidase I (COI) gene, the primer pair C1J2186 (Simon et al., 1994) and C1N2608 (Damgaard et al., 2000) was used. In some cases, when the above COI pair of primers did not yield a well-defined product, overlapping internal primers were used (COI-ScF2 5′-TTTTGATTTTTTGACATCCTGA-3′, COI-ScR1 5′-TCCTACTGTAAATATATGGTG-3′). To amplify a fragment of the 18S ribosomal gene, the primer pair 18S-ScF1 5′-CCTGTCGGTGTAACTGGCAT-3′, 18S-ScR 5′-GCTGGCTGGCATCGTTTATG-3′ was used.

Polymerase chain reaction (PCR) amplification for all DNA fragments that were analyzed was carried out in a final volume of 20 μL containing 30 ng of DNA, 1.25 U Perpetual OptiTaq (EURx, Gdańsk, Poland), 0.4 μL of 20 μM of each primer, two μL of 10× Pol Buffer B and 0.8 μL of five mM dNTPs in a Mastercycler ep system (Eppendorf, Hamburg, Germany). The cycling profile for the PCR was: 95 °C for 2 min, 35 cycles of 95 °C for 30 s, 45 °C for 30 s, 72 °C for 1 min and a final extension period of 72 °C for 7 min.

In order to assess the quality of the amplification, the PCR products were electrophoresed in 1% agarose gel for 45 min at 85 V with a DNA molecular weight marker (Mass Ruler Low Range DNA Ladder; Thermo-Scientific, Waltham, MA, USA). The PCR products were purified using Exo-BAP (EURx, Gdańsk, Poland).

Samples were sequenced in both directions using the same primers as for the PCR reactions combined with a BigDye Terminator 3.1 Cycle Sequencing Kit (Applied Biosystems, (ABI) Foster City, CA, USA) using the chain termination reaction method (Sanger, Nicklen & Coulson, 1977). The sequencing reaction was carried out with the PCR product at a total volume of 20 μL containing two μL of BigDye Terminator Reaction Ready Mix v. 3. 1 (ABI), two μL 5 × sequencing buffer (ABI), 3.2 mol/μL of the primer solution and six μL of the purified PCR product. The cycle-sequencing profile was 3 min at 94 °C followed by 30 cycles of 10 s at 96 °C, 5 s at 50 °C, and 2 min at 60 °C. Sequencing products were precipitated using ExTerminator (A&A Biotechnology, Gdynia, Poland), and were separated on an ABI PRISM 377 DNA Sequencer (Applied Biosystems, Foster City, CA, USA).

Sequence edition and alignment

Raw chromatograms were evaluated and corrected in Geneious v10. 2. 6 (https://www.geneious.com). In order to identify the numts (Bensasson et al., 2001; Song et al., 2008), the mitochondrial COI sequences were translated into amino acid sequences with Geneious v10.2.6 using the standard invertebrate mitochondrial genetic code. All of the nucleotide sequences were verified using BLAST searches of NCBI (http://blast.ncbi.nlm.nih.gov/Blast.cgi). The alignment of the sequences that were studied was performed using the MAFFT (Katoh et al., 2002) plugin within Geneious v10.2.6.

Divergence time estimate

We used Praecoris dominicana Poinar, 1991 (Holoptilinae) from Dominican amber that is estimated to be about 15–20 million years old (Iturralde-Vincent & MacPhee, 1996) as a minimum age constraint. The root was calibrated at 180 Mya based on Ceresopsis costalis Becker-Migdisova, 1958 fossil following Johnson et al. (2018). We applied a lognormal distribution and set a minimum hard bound age by applying an offset value of 15 million years. The secondary calibrating point (MRCA of Holoptilinae and Phymitinae) was selected from Masonick et al. (2017).

Divergence times and their confidence intervals were estimated using the Bayesian Markov chain Monte Carlo (MCMC) coalescent method, which was implemented in BEAST version 10.0.1 (Suchard et al., 2018). The unlinked GTR model of nucleotide substitution, gamma-distributed rate variation and a proportion of invariant sites of heterogeneity model were applied with their base frequencies estimated during the analysis. A relaxed molecular clock using the uncorrelated lognormal model was applied with a Yule process speciation prior for the branching rates. The final analysis consisted of two independent MCMC chains for 30 million generations sampling every 1,000 generations. The convergence to stationery distribution and the effective population size of the model parameters were checked using Tracer v1.7.1 (Rambaut et al., 2014). The maximum clade-credibility trees were built using TreeAnnotator v1.10.1 (Rambaut & Drummond, 2002–2018) the initial 10% of samples were discarded as burn-in. FigTree v1.4.3 (http://tree.bio.ed.ac.uk/software/figtree/) was used to visualize the result. The tree was edited and annotated in Corel Draw 17.1.0.572, 2014 Corel Corporation.

Morphological data

Taxon sampling and outgroup

A total of 32 species were included into the dataset with 27 species of the genus Sirthenea. Material in total of over 400 dry specimens was examined for this study. The taxon sampling covered all of the known Sirthenea species that are distributed in almost all of the zoogeographical regions: the Afrotropical region (including Madagascar): S. africana Distant, 1903, S. flaviceps (Signoret, 1860), S. picescens Reuter, 1887, S. rodhaini Schouteden, 1913; the Palearctic and Oriental regions: S. caiana Chłond, 2008, S. dimidiata Horváth, 1911, S. flavipes (Stål, 1855), S. kali Chłond, 2018, S. nigronitens (Miller, 1958), S. nitida Chłond, 2008, S. nigra Cai & Tomokuni, 2004, S. setosa Chłond, 2018; the Australian region: S. laevicollis Horváth, 1909, S. obscura (Stål, 1866); the Neotropical region: S. amazona Stål, 1966, S. atra Willemse 1985, S. dubia Willemse, 1985, S. ferdinandi Willemse 1985, S. jamaicensis Willemse, 1985, S. ocularis Horváth, 1909, S. pedestris Horváth, 1909, S. peruviana Drake & Haris, 1945, S. plagiata Horváth, 1909, S. stria (Fabricius, 1794), S. venezolana Maldonado Capriles, 1955, S. vidua Horváth, 1909, S. vittata Distant, 1902. The number of Sirthenea species from the New World followed Willemse (1985), the Old World species number followed Chłond (2018). Five species were used as the outgroup taxa; among these, four belonging to the subfamily Peiratinae. Several species of Peiratinae were additionally examined, although the molecular material of those specimens could not be used for the analyses. The taxon sample of the outgroup taxa was limited by the availability of dry specimens whose genetic material could be used in the current research.

Examination and deposition of the taxa

The material is stored in 24 entomological collections. The specimens that were examined during the current studies belong to: CAU—China Agricultural University, Beijing, China; HNHM—Hungarian Natural History Museum, Budapest, Hungary; MACN—Museo Argentina de Ciencias Naturales “Bernardino Rivadavia,” Buenos Aires, Argentina; MLPA—Universidad Nacional de La Plata, Museo de la Plata, La Plata, Argentina; MMBC—Moravské Muzeum, Brno, Czech Republic; MNHN—Muséum National d’Histoire Naturelle, Paris, France; MZH—Finnish Museum of Natural History, Helsinki, Finland; NHMB—Naturhistorisches Museum, Basel, Switzerland; NHMD (formerly “ZMUC”)—Natural History Museum of Denmark, Copenhagen, Denmark; NHMUK—Natural History Museum, London, United Kingdom; NHMW—Naturhistorisches Museum Wien, Vienna, Austria; NHRS—Naturhistoriska Riksmuseet, Stockholm, Sweden; NMEG—Naturkundemuseum, Erfurt, Germany; NMPC—National Museum, Prague, Czech Republic; NSMT—National Science Museum, Tokyo, Japan; RBINS—Royal Belgian Institute of Natural Sciences; RMCA—Musée Royal de l’Afrique Centrale, Tervuren, Belgium; RMNH—Naturalis Biodiversity Centre, Leiden, The Netherlands; SMF—Forschungsinstitut und Naturmuseum Senckenberg, Frankfurt am Main, Germany; TLMF—Tiroler Landesmuseum Ferdinandeum, Innsbruck, Austria; USMB—Upper Silesian Museum, Bytom, Poland; USNM—National Museum of Natural History, Washington DC, USA; ZJPC—Zdeněk Jindra private collection, Prague, Czech Republic; ZMHB—Museum für Naturkunde der Humboldt-Universität, Berlin, Germany.

Microscopy and illustrations

The characters were examined using a Nikon NiU compound microscope and photographed using a Nikon DS-Fi2 camera. Pictures of the morphological details were taken with NIS-Elements D 4.50.00 64-Bit. Each specimen was imaged from different focal planes via Z-series acquisition and automatically aligned and layered. The drawings were made freehand on a Nikon Eclipse Ni using a camera lucida. The characters of the individuals for the SEM images, which had been sputter-coated with gold in a Pelco SC-6 sputter coater, were imaged using a Phenom XL scanning electron microscope (Phenom-World B.V., Eindhoven, The Netherlands) in low vacuum conditions at a 10, 15 and 20 accelerating voltage using a secondary electron detector. The plates were prepared using Corel Draw 17.1.0.572, 2014 Corel Corporation.

Morphological characters

Since there is no morphology-based phylogenetic analysis targeting the species level relationships within the genus Sirthenea, the first morphological matrix was specifically developed for this study. The morphological characters were scored for both sexes of each species from the different localities of each species (from one to 70 individuals). A total of 64 characters were used for the analyses. A matrix of 64 characters (numbered 1–64) across the 32 terminal taxa was prepared using Mesquite version 3.5 (Maddison & Maddison, 2015). Unknown character states were coded with a “?” and inapplicable states were coded with “–”. A csv file containing the character matrix is provided in Table S2.

Description, documentation and discussion of the characters (Figs. 1–3) that were used in the analysis as well as the states and their distribution among the analyzed taxa refer to the current analysis and are provided below.

Figure 1 Morphological characters.

(A) Sirthenea nigronitens (Miller, 1958), head and thorax; (B) Calistocoris virgo Reuter, 1881, head; (C) S. picescens Reuter, 1887, head; (D) S. amazona Stål, 1866, head; (E) S. amazona Stål, 1866, pronotum; (F) Androclus granulatus Stål, 1863, head; (G) S. picescens Reuter, 1887, pronotum; (H) S. laevicollis Horváth, 1909, head; (I) S. caiana Chłond, 2008, head and pronotum; (J) Platymeris rhadamanthus Gerstaecker, 1873, head and thorax.

Figure 2 Morphological characters.

(A) Platymeris rhadamanthus Gerstaecker, 1873, head and thorax; (B) Sirthenea peruviana Drake & Harris, 1945, thorax; (C) S. laevicollis Horváth, 1909, fossula spongiosa on middle tibia; (D) S. laevicollis Horváth, 1909, structure of fossula spongiosa; (E) S. flaviceps (Signoret, 1860), fossula spongiosa on fore tibia; (F) Ectomocoris ululans (Rossi, 1807), fossula spongiosa on fore tibia; (G) E. ululans (Rossi, 1807), head; (H) E. ululans (Rossi, 1807), hemelytron; (I) S. flaviceps (Signoret, 1860), hemelytron; (J) S. ferdinandi Willemse, 1985, scutellum; (K) S. flavipes (Stål, 1855), scutellum; (L) S. pedestris Horváth, 1909, hemelytron; (M) S. flavipes (Stål, 1855), abdomen; (N) S. laevicollis Horváth, 1909, abdomen; (O) S. nitida Chłond, 2008, abdomen.

Figure 3 Morphological characters.

(A) S. pedestris Horváth, 1909, male pygophore; (B) Peirates strepitans Rambur, 1839, additional process on VII abdominal sternite of male and pygophore; (C) S. dimidiata Horváth, 1911, reduced hemelytron; (D) S. dimidiata Horváth, 1911, right paramere; (E) S. dimidiata Horváth, 1911, left paramere; (F) S. setosa Chłond, 2018, right paramere; (G) S. setosa Chłond, 2018, left paramere; (H) Platymeris rhadamanthus Gerstaecker, 1873, right paramere; (I) Platymeris rhadamanthus Gerstaecker, 1873, left paramere; (J) S. dimidiata Horváth, 1911, median process of pygophore; (K) S. setosa Chłond, 2018, median process of pygophore; (L) S. nigra Cai & Tomokuni, 2004, median process of pygophore.

Clypeus: (0) straight, directed horizontally (Fig. 1A); (1) distinctly directed downward (Fig. 1B). The apex of the clypeus in most of the studied taxa is directed downward and the state of this character was also seen in all of the outgroup taxa. Among all of the species of Sirthenea, only the Afrotropical species had a straight, horizontally directed clypeus. Both character states were observed in the representatives of Oriental and Neotropical regions. The second state of the character occurred in two species that are distributed in Australia.

Point of antennal insertion: (0) in half of the anteocular part of the head (Fig. 1A); (1) near the anterior margin of an eye (Fig. 1B). This character was represented by the second state only in all of the outgroup taxa and Sirthenea ocularis. S. ocularis was the only known species of the genus with strongly enlarged eyes, and as a result, the placement of point of antennal insertion was close to their anterior margin.

Apices of antennifers orientation: (0) directed fronto-dorsally (Fig. 1A); (1) directed fronto-ventrally (Fig. 1B). The apices of the antennifers are directed fronto-dorsally in all of the representatives of Sirthenea, and this state of character was also found in more than half of the representatives of the outgroup taxa.

Dorsal surface of antennifers: (0) smooth; (1) with a distinct sculpturation (corrugated and pointed) (Fig. 1C). Sculpturation on the dorsal surface of the antennifers is a character that was present in all of the known Oriental and Australian species of Sirthenea as well as in the S. picescens that is distributed in Madagascar. This character is also present in some Oriental species and in some of the outgroup taxa.

Mandibular plates: (0) small (Fig. 1B); (1) large (Fig. 1J). The mandibular plates of all of the Sirthenea species as well as in all of the studied taxa belonging to the subfamily Peiratinae (except Androclus Stål, 1863) were relatively small. Distinctly enlarged mandibular plates, compared with those seen in representatives of Sirthenea, were present in the outgroup genera Androclus and Platymeris Laporte, 1833.

Shape of scape: (0) club shaped (Fig. 1C); (1) slender (Fig. 1B). The first state of the character was present in most of the representatives of Sirthenea. An elongate and slender scape was characteristic only for four species of Sirthenea: the Australian species as well as the Madagascan S. picescens and the Oriental S. setosa as well as most of the outgroup taxa.

Length of scape: (0) short, not surpassing the apex of the head; (1) long, surpassing the apex of the head (Fig. 1C). Among the representatives of Sirthenea, the second state of the character was present in only three species: S. picescens, S. laevicollis and S. jamaicensis. An elongate scape, surpassing the apex of the head was a diagnostic character to support the placement of the first two above-mentioned species in a distinct subgenus Monogmus Horváth, 1909.

Length of pedicel: (0) shorter than the head; (1) equal or longer than the head (Fig. 1B). A short pedicel occurred in all of the Sirthenea species and in only four of the outgroup taxa.

Length of labial segment II: (0) short (distinctly shorter than labial segment III) (Fig. 1A); (1) long (almost the same length as labial segment III) (Fig. 1J). Short labial segment II is a state that was present in all of the representatives of Sirthenea as well as in most of the outgroup taxa. The length of labial segment II was equal or almost equal to labial segment III only in Platymeris rhadamanthus Gerstaecker, 1873.

Position of the apex of labial segment III: (0) not reaching the anterior margin of the anterior pronotal lobe of the pronotum (Fig. 1A); (1) reaching the anterior margin of the anterior pronotal lobe of the pronotum (Fig. 2G). The first state of this character was observed in all of the Sirthenea species except S. setosa. The condition of this character in S. setosa (labial segment III reaching the anterior margin of the pronotum) was a result of the segment length.

Condition of basal part of labial segment III: (0) not expanded (Fig. 1A); (1) expanded (Fig. 2G). The second state of the character was present in all of the Afrotropical and Australian species. Only three of the species distributed in Oriental region (S. nigronitens, S. nigra and S. setosa) and most of the Neotropical species did not have an expanded basal part of the segment.

Ventral surface of basal part of labial segment III: (0) not flattened (Fig. 1D); (1) flattened (Fig. 1J). The ventral surface of the basal part of labial segment III in all of the Sirthenea and most of the outgroup taxa was not flattened. Only six of the studied outgroup taxa had a distinctly flattened basal part of the ventral surface of labial segment II.

Shape of labial segment IV: (0) thin (Fig. 1A); (1) robust (Fig. 1J). The first state of the character occurred in all of the Peiratinae and it was common in most of the Reduviidae. The state (1) was an autapomorphy for Platymeris rhadamanthus.

Size of an eye in lateral view: (0) not reaching the dorsal and ventral ridges of the head (Fig. 1B); (1) reaching the dorsal and ventral ridges of the head (Fig. 1D).

Eye width: (0) narrower than the synthlipsis (Fig. 1C); (1) same width or wider than the synthlipsis. The width of an eye was evaluated in the dorsal view. The first state was present in the Sirthenea species that are distributed in Afrotropical region and most of the species from the Oriental region (except S. nigra and S. setosa). Most of the Neotropical species had the second state of this character.

Shape of posterior margin of an eye: (0) S-shape (Fig. 1A); (1) U-shape (Fig. 2G); (2) rounded (without a cavity) (Fig. 1B). To code two different shapes of the posterior margin of an eye in the lateral view as well the absence of a cavity of the margin, we treated them as one multistate character.

Ventral margin of the head: (0) convex (Fig. 1A); (1) straight or concave (Fig. 1D). The ventral margin of the head was convex in most species of Sirthenea (especially in the species that are distributed in South-East Asia and Australia) as well as in most of the outgroup taxa.

Posteroventral part of the head: (0) at same level as the ventral ridge of the neck-like part of the head; (1) lower than the ventral ridge of the neck-like part of the head (Fig. 1D). The first state of this character was present in most of the Sirthenea species (every Afrotropical species) as well as in the outgroup taxa.

Apodeme depression on the head: (0) shallow (Fig. 1C); (1) deep (Fig. 1H). Among the representatives of Sirthenea only two species S. setosa (Oriental region) and S. laevicollis (Australian region) had a deep apodeme depression on the head, while it was common character in the outgroup taxa.

Degree of the depression of transversal furrow of the head: (0) shallow (Fig. 1C); deep (1) (Fig. 1H). A shallow transversal furrow of the head was present in the Afrotropical and some of the Oriental representatives of Sirthenea. A deep furrow was characteristic for the Australian and Neotropical (except S. jamaicensis) species.

Shape of the transversal furrow of the head: (0) triangular (Fig. 1I); (1) rounded (Fig. 1C). The shape of the transversal furrow of the head was unified (triangular) only in the Afrotropical species of Sirthenea.

Ocelli: (0) present (Fig. 1A); (1) absent. Lack of ocelli only occurred in one species, S. kali, which is known only from brachypterous females. Shortened wings are often connected with the absence of ocelli, however, the ocelli that were present in the brachypterous females that were found in three other species: S. rodhaini, S. dimidiata and S. laevicollis. This character was an apomorphy for S. kali.

Postocular part of the head: (0) equal to or lower than the anteocular part in the lateral view; (1) distinctly elevated compared to the anteocular part in the lateral view (Fig. 1D). This character was coded as (0) only in the African representatives of Sirthenea. The second state of the character was uniform in the representatives of the Australian region.

Postocular enlargement of the head: (0) absent; (1) present (Fig. 1F). Among the Sirthenea only S. jamaicensis had an enlargement behind the posterior margin of an eye, however, it was a characteristic structure for most of the outgroup Peiratinae.

Lateral processes on the neck-like part of the head: (0) absent; present (1) (Fig. 1F). This character was coded (1) in most of the outgroup taxa that belong to Peiratinae. This structure was absent in the representatives of Sirthenea.

Length of the anterior pronotal lobe of pronotum: (0) longer than the posterior pronotal lobe of the pronotum (Fig. 1I); (1) shorter than the posterior pronotal lobe of the pronotum (Fig. 1J). The second state of this character was present only in Platymeris rhadamanthus.

Width of the anterior pronotal lobe of pronotum in males: (0) similar width as the posterior pronotal lobe of the pronotum; (1) distinctly narrower than the posterior pronotal lobe of the pronotum (Fig. 1I). This character was based on the measurements of the anterior and posterior lobes of the pronotum and it was uniformly coded (as 1) only in the Australian representatives of the genus. In line with the findings or previous taxonomic studies, the females of Sirthenea were larger than the males and the width of their anterior lobe of the pronotum was similar to the width of the posterior lobe.

Collar of pronotum: (0) reduced; (1) distinctly separated (Fig. 1G). Our studies recovered two different states of this character in Sirthenea. A distinct collar was present in all of the Neotropical and Australian representatives of the genus. Two states of this character were recognized as being a diagnostic feature of the Afrotropical and Oriental species of Sirthenea.

Anterolateral angles of collar of pronotum: (0) not enlarged; (1) distinctly enlarged (Fig. 1G). The enlargements of the anterolateral angles of the pronotum in the Australian and most of the Neotropical species of Sirthenea as well as in almost all of the outgroup taxa were very distinct.

Sulci of anterior the pronotal lobe: (0) distinct (Fig. 1I); (1) reduced (Fig. 1G). The second state of the character was present only in two species: S. picescens and S. laevicollis. A reduced sulci of the anterior lobe of the pronotum was the second diagnostic character to support the placement of the first two above-mentioned species in the subgenus Monogmus and it was also present in three other species from the outgroup taxa belonging to the subfamily Peiratinae.

Shape and length of apodeme depression of anterior pronotal lobe: (0) long and thin (Fig. 1G); (1) short and wide (Fig. 1I). Most of Sirthenea species possess long and thin apodeme depression of pronotum.

Anterior pronotal lobe spinelike processes: (0); (1) present (Fig. 1J). This character was present in only one outgroup taxon—Platymeris rhadamanthus.

Shape of transversal furrow of the pronotum latero-external sulci: (0) not curved; (1) curved (Fig. 1I). A curved transversal furrow in the place of connection with latero-external sulci occurred in many species of Sirthenea that were distributed in various zoogeographical regions.

Posterior pronotal lobe: (0) smooth; (1) distinctly sculptured or pointed (Fig. 1I). In most species of Sirthenea, the posterior pronotal lobe was smooth as opposed to the outgroup taxa where most of the species had a distinct sculpture on this structure.

Posterior margin of the posterior pronotal lobe: (0) without a cavity (Fig. 1I); (1) with a distinct cavity (Fig. 1E). This character is coded as (0) for all of the studied species except for S. amazona, which had a unique, very distinct cavity.

Spine-like processes on the postolateral angles of the posterior pronotal lobe of the pronotum: (0) absent; (1) present (Fig. 2A). This character was similar to character 32 and was also present exclusively in Platymeris rhadamanthus.

Stridulitrum: (0) not elongated anteriorly; (1) elongated anteriorly (Fig. 1J). An anteriorly elongated stridulitrum was present only in Platymeris rhadamanthus; this state was absent in all of the studied genera of Peiratinae.

Prosternal process: (0) elongated (Fig. 1A); (1) short. The prosternal process in all of the studied Peiratinae was long and distinctly surpassed the fore coxae. Only in Platymeris rhadamanthus was it relatively short and did not surpass the posterior margin of the procoxal cavity.

Ridges of the proepisternum and proepimeron: (0) connected along the entire length (Fig. 1J); (1) not connected along the entire length (Fig. 1A). This character was coded as (0) in all of the African, Australian and most of the Neotropical species of Sirthenea. The same state of character was observed in most of the outgroup taxa.

Posterior angles of the dorsolateral part of metapleura: (0) arcuate; (1) bent at a right angle (Fig. 2B); (2) depressed (Fig. 1J). This multistate character was coded as (2) only in Platymeris rhadamanthus. Arcuate angles of the metapleura were present in all of the African representatives of Sirthenea as opposed to the state of the same character in almost all of the species from the New World (except S. jamaicensis).

Metapleura: (0) with two complete ridges; (1) with one complete ridge (Fig. 2B). This character was coded (1) for all of the species of Sirthenea from Australian and Neotropical regions as well as most of the Afrotropical and Oriental regions. Only three species of outgroup taxa had two complete ridges of the metapleura.

Fore coxae: (0) elongated (Fig. 1A); (1) not elongated. This feature coded as (0) in all of the Peiratinae and was used as one of the main diagnostic characters of this subfamily.

Presence of the fossula spongiosa on the middle tibia: (0) absent; (1) present (Figs. 2C and 2D). The fossula spongiosa is a hairy structure on the tibiae that occurs in many Reduviidae. This structure shows a substantial homoplasy within the Reduviinae and it was treated as a plesiomorphic feature. Zhang et al. (2016b) found evidence for a single origin of this structure in Reduvioidea with numerous reductions. Among all of the species of Sirthenea, this feature was present only on the middle tibia of S. laevicollis. The fossula spongiosa on the middle tibia was also present in all of the outgroup taxa.

Length of the fossula spongiosa on the fore tibia: (0) longer than half of length of tibia (Fig. 2F); (1) equal or shorter than half of length of tibia (Fig. 2E). Only three species of Sirthenea had a short fossula spongiosa on fore tibia.

Brachyptery in females: (0) absent; (1) present (Fig. 3C). The previous study documented brachyptery in the genus Sirthenea for the first time. Brachypterous females were found in six species: S. rodhaini, S. dimidiata, S. kali, S. laevicollis, S. jamaicensis and S. vidua; the last four species are known exclusively from brachypterous females. Among the outgroup taxa, only the females of Ectomocoris ululans (Rossi, 1807) were brachypterous.

Basal part of the clavus: (0) median longitudinal groove absent; (1) median longitudinal groove present (Fig. 2J). This character was coded (1) in the S. kali from the Oriental region; however, a longitudinal groove on the clavus was present in most of the New World species of Sirthenea.

Depression on the basal half of the clavus: (0) absent; (1) present (Fig. 2L). This character was present in a few species of Sirthenea.

Width of the clavus: (0) equal to the cell formed by the cubital and postcubital veins (Fig. 2A); (1) distinctly wider than the cell formed by the cubital and postcubital veins (Fig. 2H). The second state of this character was found only in Ectomocoris ululans.

Apical internal cell surface: (0) less than half of the surface of the apical external cell (Fig. 2I); (1) more than half of the surface of the apical external cell (Fig. 2H). A small apical internal cell was observed in all of the Sirthenea species as opposed to the outgroup taxa, in which only Calistocoris virgo (Miller, 1940) had a small apical internal cell, and the state of this feature (1) was dominant.

Shape of the apical internal cell: (0) quadrilateral (Fig. 2I); pentagonal (Fig. 2H). The pentagonal apical internal cell was present only in S. laevicollis and four species of the outgroup taxa.

Cu vein: (0) curved (Fig. 2H); (1) straight (Fig. 2I). The vein was straight in only four representatives of Sirthenea (distributed in the Oriental and Neotropical regions).

Length of the apex of the scutellum: (0) short (Fig. 2K); (1) elongated (Fig. 2J). An elongated apex of scutellum was present in all of the Oriental, Australian and Neotropical species of Sirthenea as well as in S. picescens (Afrotropical region) and S. setosa (Oriental region). Only three taxa from the outgroup had a short apex of the scutellum.

Apex of the scutellum: (0) directed horizontally; (1) directed vertically (Fig. 1J). The only representatives with a vertically directed apex of the scutellum were the species from the outgroup taxa.

Depression on the scutellum: (0) triangular (Fig. 2K); (1) tongue-like (Fig. 2J). This character was coded (1) in most of the analyzed taxa. Among the Sirthenea, only a few species from Afrotropical and Oriental regions had a triangular depression on the scutellum.

Lateral spine-like processes of the scutellum: (0) absent; (1) present (Fig. 2A). This character was present only in Platymeris rhadamanthus. This feature was common in representatives of the subfamily Reduviinae.

Lateral angles of the basal part of the scutellum: (0) rounded (Fig. 2K); tightened (Fig. 2J). This feature was coded (0) in all of the Afrotropical and Oriental species of Sirthenea with the exception of S. caiana. The species that are distributed in the Australian and Neotropical regions had tightened lateral angles of the scutellum.

Lateral depressions of the basal part of the scutellum: (0) shallow (Fig. 2K); (1) deep (Fig. 2J). Shallow lateral depressions of the scutellum were characteristic only for the Afrotropical species of Sirthenea and S. amazona.

Connexives orientation: (0) vertical (Fig. 1J); horizontal (Fig. 2H). The first state of this character was present in all Sirthenea species but also occurred among the coded outgroup taxon—C. virgo.

Distance of spiracles III–IV from the ventral connexival suture: (0) distant (Fig. 2M); (1) connected (Fig. 2N); (2) fused (Fig. 2O). Distance of the abdominal spiracles from the ventral connexival suture was combined into one multistate character, and it as coded as (2) only in only two representatives of Sirthenea: S. nitida and S. ocularis.

Posterior margin of the VIII abdominal sterna of males: (0) without an indentation (Fig. 3A); (1) with an indentation (Fig. 3B). The Sirthenea species that are distributed in the Afrotropical, Oriental (except S. kali and S. nigronitens, which are known only from females) and the Australian regions had a distinct indentation on the VIII abdominal sterna of males. Neotropical species of the genus were devoid of such a structure.

Parameres: (0) symmetric (Figs. 3H and 3I); (1) asymmetric (Figs. 3D–3G). All of the representatives belonging to the subfamily Peiratinae were characterized by asymmetric parameres.

Shape of the parameres: (0) club-shaped (Figs. 3D and 3E); (1) wide (Figs. 3F and 3G); (2) narrow (Figs. 3H and 3I). This multistate character was coded (2) only in the case of Platymeris rhadamanthus. Among the Sirthenea, most of the species had very characteristic, club-shaped parameres. The only exceptions were both of the species that are distributed in Australia: S. laevicollis and S. obscura as well as the Oriental S. setosa and Madagascan S. picescens.

Additional process on the VII abdominal sterna of males: (0) absent; (1) present (Fig. 3B). This character was present in only one outgroup taxon—Peirates strepitans Rambur, 1839.

Median process of the pygophore: (0) short (Fig. 3J); (1) long (Fig. 3K); (2) bulbous (Fig. 3L). A long median process of the pygophore was present in all of the outgroup taxa belonging to the subfamily Peiratinae and in both species that are distributed in Australia: S. laevicollis and S. obscura as well as the Oriental S. setosa and the Madagascan S. picescens. This species had a presence that was similar to feature 62. The only species with a bulbous median process of the pygophore was S. nigra.

Phylogenetic analyses

A combined matrix of the molecular (939 bp) and morphological (64 characters) data for the total number of taxa under study (32) was analyzed using the Bayesian Inference (BI) method. Separate analyses of molecular and morphological datasets were also performed. Gaps were treated as missing data in all of analyses.

The alignment was initially partitioned by gene and for protein-encoding genes, by position. The optimal partitioning scheme and the corresponding models of nucleotide evolution were determined using PartitionFinder v. 2.1.1 (Lanfear et al., 2016) using the Bayesian Information Criterion. Only models for “mrbayes” were considered, the branch lengths were unlinked and the search was set to the “greedy” algorithm (Lanfear et al., 2012). The morphological data in the combined matrix were analyzed as a single, separate partition using the maximum likelihood model for the discrete morphological character data under the assumption that only characters that varied among the taxa were included (Mkv) (Lewis, 2001). Bayesian analyses were performed using MrBayes v3.2.6 (Ronquist et al., 2012) running on a CIPRES Science Gateway v3.3 (phylo.org). All analyses used four chains (one cold and three heated) and two runs of 30 million generations with default prior settings. The analyses were conducted with a gamma distribution for the morphological partition. Convergence of both runs was visualized in Tracer v1.6 (Rambaut et al., 2014) as well as by an examination of the potential scale reduction factor (PSRF) values and average standard deviation of the split frequencies in the MrBayes output.

Trees were edited and annotated in Corel Draw 17.1.0.572, 2014 Corel Corporation.

Nodes with (BI) posterior probability (PP) > 0.95 were considered to be strongly supported, with PP = 0.90–0.94 moderately supported, and with PP = 0.85–0.89 (0.80–0.89 in morphology-based analysis) weakly supported. Nodes with PP < 0.85 (or 0.80) were considered to be unsupported.

Results

Molecular data analyses

Divergence time estimate

The age estimates (median age values and 95% Highest Posterior Density (HPD)) from the BEAST analysis are presented in Fig. 4. According to the relaxed molecular clock analysis of the molecular data, the genus Sirthenea started diverging in the Late Cretaceous (74,83 Mya, 95% HPD 109,3–43,8 Mya) into two clades. The clades began to branch in the Paleocene. The first clade diverged 65,87 Mya (node A, 95% HPD 98–38,5 Mya) and comprises S. nigra, S. dimidiata, S. flaviceps, S. caiana, S. rodhaini, S. flavipes, S. picescens and S. africana. The second clade diverged 62,1 Mya (node B, 95% HPD 93,6–35,5 Mya) and is formed by S. atra, S. obscura, S. nitida, S. peruviana, S. pedestris, S. kali, S. laevicollis, S. setosa, S. vittata, S. ocularis, S. nigronitens, S. jamaicensis, S. dubia, S. ferdinandi, S. plagiata, S. vidua, S. venezolana, S. stria and S. amazona. Together with species from the genus Sirthenea, four species from other genera from the subfamily Peiratinae and Platymeris rhadamanthus from the subfamily Reduviinae also branched. Most species of the investigated genus came into existence in the Neogene.

Figure 4 Estimated ages of divergencefor the Sirthenea genus created using BEAST.

Branch lengths are drawn proportional to time. A total of 95% highest posterior density intervals for nodes are indicated by horizontal blue bars. Letters A and B indicates first and second clade respectively. Taxa are highlighted by their biogeography.

Phylogenetic analyses

The BI analysis reached convergence with the average standard deviation of split frequencies that was well below 0.01. The mixing of the MCMC chains was good since the effective sample sizes (ESS) for the PP were much higher than 200 and the observed PSRF was 1.00. The results of molecular datasets analyzed separately are shown in Figs. 5 and 6, respectively, while the tree resulting from the combined analysis is shown in Fig. 7.

Figure 5 A total of 50% majority rule consensus tree from a Bayesian analysis of morphological dataset.

Posterior probabilities are shown at the respective nodes.

Figure 6 A total of 50% majority rule consensus tree from a partitioned Bayesian analysis of molecular dataset.

Posterior probabilities are shown at the respective nodes.

Figure 7 A total of 50% majority rule consensus tree from a partitioned Bayesian analysis of a combined data set (DNA and morphology).

Posterior probabilities and jackknife values are shown at the respective nodes separated by a forward slash “/.” If a node was unsup.

The total evidence analysis (Fig. 7) recovered the genus Sirthenea polyphyletic with all of the outgroup genera, except for Platymeris and Androclus resolved within one clade. The position of the Sirthenea africana + Androclus granulatus Stål, 1863 clade was unresolved. All other Sirthenea species and representatives of Peirates Serville, 1831, Calistocoris Reuter, 1881, and Ectomocoris Mayr, 1865 were resolved together, although without a support and consisted of two well supported clades. The first clade consists of Sirthenea flavipes, S. flaviceps, S. rodhaini, S. caiana, S. nigra and S. dimidiata (PP = 1) with Peirates strepitans recovered as a sister with a strong support (PP = 0.99). All of the other Sirthenea species formed a monophylum along with the species of the genera Calistocoris, and Ectomocoris with strong support (PP = 1). Within this clade, the Sirthenea picescens and Ectomocoris ululans species branched off first. Next, the S. amazona was recovered to be sister to S. stria + S. venezolana clade (PP = 0.97) and all together sister to a very well-supported monophylum consisting of the remaining species of this clade and species of Calistocoris (PP = 0.99). In this clade, Calistocoris virgo was recovered sister to the Sirthenea species with strong support (PP = 1).

The results of analyses of the morphological (Fig. 5) and molecular (Fig. 6) data separately were not congruent and the major difference was the monophyly of Sirthenea. While the results of molecular-based analysis were largely congruent with the results of the combined analysis, the morphology-only analysis recovered Sirthenea as monophyletic with low support (0.82). All of the outgroup genera, except for Calistocoris were unresolved, while Calistocoris virgo was recovered as sister to Sirthenea without a support.

Discussion

Our results are the first comprehensive analysis concerning one of the representatives of the subfamily Peiratinae. A phylogeny and molecular analysis of the genus Sirthenea that is based on a worldwide sampling has been missing to date. The morphological and molecular analyses of 27 known species of Sirthenea and five outgroup taxa (four Peiratinae and one Reduviinae) allowed us to integrate the phylogenetic relationships and the Sirthenea time tree with the spatial history in the context of palaeogeography (Sanmartin & Ronquist, 2004; McIntyre et al., 2017). We also verified a correlation of the divergence dates (Fig. 4) based on the molecular clocks of all known Sirthenea species that belong to the geographically separated taxa. As biogeographic history is also far too complex to be inferred from ancestral reconstructions alone, our interpretations are derived from the joint molecular dating, morphological analysis and the phylogenetic patterns within clades (Figs. 1–7) as well as specific geological periods. Our results suggest that the evolution of Sirthenea is considerably complex and cannot be limited to a sequence of dispersal and vicariance events. However, because the diversity of the genus and the young age of some of the species (Fig. 4) we cannot exclude that dispersal could also played a role in the processes that have shaped the global distribution of Sirthenea, what we have demonstrated in our previous studies (Chłond, Bugaj-Nawrocka & Sawka-Gądek, 2017). The molecular clock data indicate that most of the known species are dated at the beginning of Paleogene 74–61 Mya (Fig. 4). The clade A comprising species S. africana, S. caiana, S. dimidiata, S. flavipes, S. nigra, S. flaviceps, S. picescens, S. rodhaini, as well as the clade B: S. kali, S. nigronitens, S. nitida, S. setosa, S. laevicollis, S. obscura, S. amazona, S. atra, S. dubia, S. ferdinandi, S. jamaicensis, S. ocularis, S. pedestris, S. peruviana, S. plagiata, S. stria, S. venezolana, S. vidua and S. vittata (Fig. 4) diverged about 75 Mya. Our results show that S. picescens is one of the oldest species in the genus. Conversely, the molecular data provide evidence that the youngest species is S. flavipes (9,17 Mya), which, can be treated as a model species, thus partially confirming the role of dispersion and allopatric speciation within the representatives of Sirthenea (Chłond, Bugaj-Nawrocka & Sawka-Gądek, 2017). Molecular clock also revealed that the main radiation of Sirthenea lasted from Eocene to the mid-Miocene, 46–13 Mya (Fig. 4). The incongruence between the results of molecular (including combined) (Figs. 6 and 7) and morphological analyses (Fig. 5) is not uncommon issue (Wieczorek et al., 2017; Tang et al., 2018) as both data source often suggest conflicting results. We carefully evaluated a possibility of either splitting Sirthenea into several genera, or including the other genera into our studied genus. Both possibilities, though, seem to be premature as the genus is clearly diagnosable by several morphological synapomorphies, for example, lack of fossula spongiosa on middle tibiae, orientation of apices of antenniferes, distinctly elongated head (mostly anteocular part), shape of the head, orientation of apical part of the head. We hypothesize that the resulting polyphyly recovered by molecular data analyses are a result of limited gene sampling rather that true non-monophyly. Undoubtedly, this calls for further studies and addition of other gene markers is planned for the near future. Nevertheless, the obtained results encouraged us to discuss possible scenarios of evolutionary history of the particular lineages currently classified as the genus Sirthenea.

As we did not find any other known worldwide distributed genus among Peiratinae (Chłond & Bugaj-Nawrocka, 2015), there is a possibility, which is supported by geological, bioclimatic and environmental events (Couvreur, Forest & Baker, 2011; Baker & Couvreur, 2013; McIntyre et al., 2017), that the distribution of Sirthenea may be the result of a change of the microhabitat it occupied. An analysis of the climatic preferences in various ecoregions (Chłond & Bugaj-Nawrocka, 2015) suggested that the representatives of the genus are mainly linked to tropical and temperate climates that have tree vegetation such as the tropical and subtropical moist broadleaf forests biomes, temperate broadleaf biomes and mixed forest biomes that are widespread in Africa, South Asia, southern North America and the northern parts of South America and Australia. A reconstruction of the biogeographic history indicated that these types of ecological niches initially diversified during the mid-Cretaceous around 100 Mya at the northern and middle latitudes on the supercontinent of Laurasia (Upchurch & Wolf, 1987; Davis et al., 2005; Couvreur, Forest & Baker, 2011; Baker & Couvreur, 2013; Couvreur & Baker, 2013). Palaeoflora that is attributable to similar niches was found in the early Paleocene of North America (Johnson & Ellis, 2002) and the late Paleocene of South America (Jaramillo, Rueda & Mora, 2006; Jaramillo et al., 2010; Wing et al., 2009) and Africa (Raven & Axelrod, 1974; Morley, 2000; Jacobs, 2004). These facts are congruent with our results: South American, Australian and part of Asian species grouped into one clade B, and Madagascan, some of the African and Asian species grouped into clade A (Fig. 4). As the origin of most Sirthenea species is dated at the beginning of Paleogene 74–61 Mya (Fig. 4) we can conclude, that the genus is of South origin, and was probably undivided until African-Madagascan-Indian plates separated from Antarctica-Australian-South American plate. The direction of dispersal was most likely from drifting Africa to Madagascar, India and continental Asia, and from South America to Australia and Asia. In this case, the molecular phylogeography of the genus is consistent with the phylogeography of other group of insects (Sklenarova, Chesters & Bocak, 2013; Ye et al., 2018) The faunal links between South America and Australia are presumably due to the long period of geological contact between those continents via Antarctica (Ye et al., 2018). The biogeographic relationships among the fauna of the southeastern tropics (Africa, Madagascar, India and Southeast Asia) are explained as being the result of a recent dispersal along the coasts of the Indian Ocean (Sklenarova, Chesters & Bocak, 2013). Exchanges between the eastern landmasses have been frequent since the collision of the Australian and Asian plates in the Miocene (Sanmartin & Ronquist, 2004; Ye et al., 2018; McIntyre et al., 2017) and the creation of Wallacea. Wallacea in its present form is a relatively young configuration of islands between Asia and Australia and the Australian and Oriental lineages could use this dispersal route since the Australian plate approached Asia 20–15 mya (Sklenarova, Chesters & Bocak, 2013; Chłond, Bugaj-Nawrocka & Sawka-Gądek, 2017). Although this route provided limited dispersal opportunities, our analysis revealed two relatively young species (Fig. 4) in this area: S. nigronitens (New Guinea) and S. setosa (Malesia—Borneo).

While evaluating the phylogeny of Sirthenea (Figs. 5 and 6; Table S1), we found that the fossula spongiosa and its development seems to be one of the most important morphological characters, most likely associated with the ecological niche that is inhabited which would support our hypothesis (Livingstone & Ambrose, 1984; Zhang et al., 2016b). The mentioned structure is a cushion-like expanded area on the tibia that is composed of thousands of minute hairs (Figs. 2C–2F), and the basic function of this structure is a prey capture (Weirauch, 2007, 2008; Zhang et al., 2016a, 2016b). We found that the fossula spongiosa is present on the fore and middle tibiae in all Peiratinae, while in Sirthenea, it is only present on fore legs—with exception of S. laevicollis, which is distributed exclusively in Australia. The development of the fossula spongiosa on the fore tibia is considered here as a reduction (in relation to other Peiratinae) and should be treated as an apomorphy for the Sirthenea. It has been suggested that these losses may be correlated with the evolution of a raptorial type of legs (Livingstone & Ambrose, 1984; Weirauch, 2007; Zhang et al., 2016b). In the non-specialized predators (in terms of prey), this type of reduction, as well as strongly enlarged fore femurs and tibiae, suggests that only the fore legs are used to grip prey, which may possibly indicate a correlation with capturing the prey from hiding. Studies on various types of raptorial legs suggest that the loss of the fossula spongiosa is not directly correlated with the evolution of alternative types of raptorial legs, but may be associated with the type of prey, predatory behavior, salivary toxicity and the morphological adaptations that pose intricate and interrelated factors that have influenced the evolution of leg structures (Zhang et al., 2016b). Moreover, comparative studies of the fossula spongiosa provide evidence that this structure has a direct impact on the efficiency of capturing prey in their ecosystems. The development of the fossula spongiosa is also related to the gradual transformation of tropical rainforests (reduction) into scrub jungles and semiarid zones (strong development) (Livingstone & Ambrose, 1984).

Tropical and subtropical moist microhabitats, with an abundance of litter fauna that provide a rich variety of prey and proper habitat conditions on the ground (leaves, elements of trees, stones), indicate the presence of a ground-dwelling predator such as Sirthenea with reduced fossula spongiosa, and preys on other insects in this particular environment. In contrast, the maximum extent of the fossula spongiosa development in other genera of Peiratinae (e.g., Ectomocoris—with an extremely developed fossula spongiosa (Fig. 2F)) is characteristic for species that live in scrub jungles and semiarid zones. A scarcity of prey in these ecosystems forced adaptations that prevent any vagrant prey from escaping the firm grip of the fossula spongiosa (Livingstone & Ambrose, 1984). The presence of this structure on the middle tibia that is only observed in S. laevicollis is probably connected with living in more arid niches (Chłond & Bugaj-Nawrocka, 2015) than those that are inhibited by other representatives of the genus. The molecular clock shows that S. laevicollis is one of the youngest species of the genus (Fig. 4), which seems to confirm that the occurrence of the fossula spongiosa on the middle tibiae is an autapomorphy for this relatively young species, whose ancestor branched off after the separation of Australia (Ye et al., 2018; McIntyre et al., 2017). The results of previous studies (Livingstone & Ambrose, 1984; Zhang et al., 2016b) seem to confirm our hypothesis about the evolutionary history of Sirthenea. The loss of the fossula spongiosa on the middle tibia was probably connected with a change and/or adaptation to newly formed, more humid ecological niches (Morley, 2000; Johnson & Ellis, 2002; Jacobs, 2004; Jaramillo, Rueda & Mora, 2006, Jaramillo et al., 2010; Couvreur, Forest & Baker, 2011; Baker & Couvreur, 2013), characterized with variety of prey and a low level of competition. This kind of niches could be colonized by different predatory insects that were able to adapt to life in such ecosystems.

Conclusions

The results of our estimations of divergence times are consistent with the results that have been obtained by other researchers (Hwang & Weirauch, 2012; Zhang et al., 2016a). The combination of the estimations of divergence times and a total-evidence analysis (present studies), as well as a model of the range of distribution (Chłond & Bugaj-Nawrocka, 2015) that is connected with the reconstructions of biogeographic history of Earth (Couvreur, Forest & Baker, 2011; Baker & Couvreur, 2013; Couvreur & Baker, 2013; Ye et al., 2018; McIntyre et al., 2017), indicate that future analyses should focus mostly on biogeographic reconstructions to support our hypothesis. Further phylogenetic studies of all of the known genera of subfamily Peiratinae are required in order to understand the relationships within subfamily (Weirauch & Munro, 2009; Hwang & Weirauch, 2012; Zhang et al., 2016a, 2016b). The present studies complement our knowledge about Sirthenea and should be considered as the beginning of the phylogenetic exploration of the genus. As its distribution (Chłond & Bugaj-Nawrocka, 2015) and taxonomy (Willemse, 1985; Chłond, 2018) are already well known and we provide the preliminary phylogeny of Sirthenea, the next step should be an increased taxon and gene sampling to resolve the questions of Sirthenea non-monophyly and its systematic position within Peiratinae.

Supplemental Information

Supplemental Information 1 Detailed list of specimens examined for themorphological data matrix with information about localities of collection.

Click here for additional data file.

Supplemental Information 2 Detailed information of specimens and GenBank accession numbers for the sequenced data.

Click here for additional data file.

Supplemental Information 3 Morphological data matrix.

Click here for additional data file.

Supplemental Information 4 18S sequences.

Click here for additional data file.

Supplemental Information 5 COI sequences.

Click here for additional data file.

The first author would like to thank to Wanzhi Cai (CAU), Dávid Rédei (HNHM), Diego José Carpintero (MACN), Maria Cecilia Melo (MLPA), Petr Baňař (MMBC), Eric Guilbert (MNHN), Anders Albrecht (MZH), Daniel Burckhardt (NHMB), Henrik Enghoff (NHMD), Mick Webb (NHMUK), Herbert Zettel (NHMW), Gunvi Lindberg (NHRS), Petr Kment (NMPC), Jerome Constant (RBINS), Eliane de Coninck (RMCA), Yvonne van Nierop (RMNH), Stephan Blank (SMF), Ernst Heiss (TLMF), Roland Dobosz (USMB), Thomas J. Henry (USNM), Zdeněk Jindra (ZJPC), Yvonne van Nierop (RMNH) and Jürgen Deckert (ZMHB) for the possibility of studying the material in the collections under their curation. We are grateful to the CIPRES Scientific Gateway that provides access to computational resources.

Additional Information and Declarations

Competing Interests

Author Contributions

DNA Deposition

Data Availability

The authors declare that they have no competing interests.

Dominik Chłond conceived and designed the experiments, performed the experiments, analyzed the data, contributed reagents/materials/analysis tools, prepared figures and/or tables, authored or reviewed drafts of the paper, approved the final draft, contibuted the specimens and founds.

Natalia Sawka-Gądek performed the experiments, analyzed the data, contributed reagents/materials/analysis tools, prepared figures and/or tables, authored or reviewed drafts of the paper, approved the final draft.

Dagmara Żyła performed the experiments, analyzed the data, contributed reagents/materials/analysis tools, prepared figures and/or tables, authored or reviewed drafts of the paper, approved the final draft.

The following information was supplied regarding the deposition of DNA sequences:

COI sequences accessible via GenBank accession numbers KX258666, MH89352–MH894388.

18S sequences accessible via GenBank accession numbers MH925974–MH926011.

The following information was supplied regarding data availability:

Raw data is available in the Supplemental Files.

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
