# Peer review of "Genetic data of museum specimens allow for inferring evolutionary history of the cosmopolitan genus Sirthenea (Heteroptera: Reduviidae)"

_PeerJ, doi:10.7717/peerj.6640_

## Round 0.1 · original submission · Minor Revisions

Dear Dr. Chlond and colleagues:

Thanks for submitting your manuscript to PeerJ. I have now received two independent reviews of your work, and as you will see, both are very favorable. Well done! Nonetheless, the reviewers raised some relatively minor concerns about the research, and areas where the manuscript can be improved. I agree with the reviewers, and thus feel that their concerns should be adequately addressed before moving forward.

Therefore, I am recommending that you revise your manuscript accordingly, taking into account all of the issues raised by the reviewers. I do believe that your manuscript will be ready for publication once these issues are addressed.

Good luck with your revision,

-joe

Reviewer 1 ·

Basic reporting

The paper reports substantial new research and analysis of a major group of true bugs with very good taxonomic diversity represented in the analyses.
The manuscript is presented in an intelligible fashion and written in standard English. The background of the performed study is clearly presented with the special emphasis of the gaps in the present knowledge of reduviids’ phylogeny and evolution. The literature cited is well-chosen. All morphological characters, presented in the data set matrix (raw data), are very well figured (Figs. 1-3) and clearly described.

Experimental design

The Chapter Material and Methods is very well organized with distinct subdivision for three main tasks: Molecular data analyses with the divergence time estimation and phylogenetic analysis of the taxa studied as well as Morphological analyses with attaching very detailed data set matrix. Therefore, the analysis can be repeated by other researchers and the matrix can be used for other studies on phylogeny of true bugs. All analysis are based on strong data obtained.

Validity of the findings

Very few reports have attempted to present phylogenetic analyses of both molecular and morphological datasets in a single paper - therefore presented manuscript is an important contribution in systematics of this group of insects. Complement in the form of a molecular clock completes the work well (Fig. 4). In the manuscript, the background of the ecological conditions (i.e. the presence of the available niches), sheds new light on Sirthenea global distribution. This aspect is thoroughly discussed. Conclusions are well stated. It is worthy to underline, that the work is well connected with earlier studies of authors on the genus Sirthenea.

Additional comments

The presented manuscript is a significant contribution to understanding the relationships within interesting assasin bugs genus of unique global distribution. It is an important study because it combines molecular, morphological and biological data with broad taxonomic sampling. The paper is nicely organized overall. The data analysis is precise, appropriate and thorough. The results and figures are clear. The discussion of the results and the implications for further study are very clearly stated.
Comment on strengths of the manuscript
- Important group of insect with a stable taxonomy (the revision of the genus Sirthenea has already been done) and unique global distribution;
- extensive data set compiled from numerous worldwide entomological collections (all known species of the genus Sirthenea have been analyzed);
- an innovative method of DNA extraction (the ability to analyze very old material);
- well defined ecological and biogeographical background to the obtained results, especially “the fossula spongiosa theory” .

Comment on weaknesses of the manuscript
- Poorly presented abstract which definitely should be improved
- Information if the Morphological matrix is an original ones, created by the authors should be added.

My only reservation is to do with the abstract improving. This part needs more information on materials and methods and results obtained. This is important work and deserves to be much better presented.

Reviewer 2 ·

Basic reporting

no comment

Experimental design

no comment

Validity of the findings

I feel that the Discussion section not sufficiently elaborates the phylogeographic aspect of the study. From the presented results it seems to me, that the genus is of South origin, and was undivided untill African-Madagascan-Indian plates separated from Antarctica-Australian-South American plate, which gives the perfect match to the molecular phylogeography of the genus. It is relatively easy to find corresponding papers and cite them and discuss the potential routes of divergence and migration. In this respect some more discussion on the potential habitat of divergence would be also appriciated.

Additional comments

I would also recommend transfer of detailed list of morphological traits studied to Supplementary Files. They are very important but very long, being only specialists intrested in, and for general public they are of lesser importance. I would keep the table with features in the text, just to give the reader the opportunity to see the general morphology of the genus. The tables may be described as they are, only with refference to supplementary material for detailed information.

---

## Round 0.2 · accepted · Accept

Dear Dr. Chlond and colleagues:

Thanks for revising your manuscript based on the concerns raised by the two reviewers. I now believe that your manuscript is suitable for publication. Congratulations! I look forward to seeing this work in print, and I anticipate it being an important resource for communities studying Sirthenea systematics and overall reduviid biology. Thanks again for choosing PeerJ to publish such important work.

Best,

-joe

#